# Investigation and Estimation of Groundwater Level Fluctuation Potential: A Case Study in the Pei-Kang River Basin and Chou-Shui River Basin of the Taiwan Mountainous Region

Nai-Chin Chen [1], Hui-Yu Wen [1], Feng-Mei Li [1], Shih-Meng Hsu [2,*], Chien-Chung Ke [1], Yen-Tsu Lin [3] and Chi-Chao Huang [3]

[1] Advanced Geological Research Task Force, Sinotech Engineering Consultants, Inc., Taipei 114065, Taiwan; ncchen@sinotech.org.tw (N.-C.C.); hywen@sinotech.org.tw (H.-Y.W.); lfm@sinotech.org.tw (F.-M.L.); ccke@sinotech.org.tw (C.-C.K.)

[2] Department of Harbor and River Engineering, National Taiwan Ocean University, Keelung 202301, Taiwan

[3] Central Geological Survey, Ministry of Economic Affairs, New Taipei City 23568, Taiwan; ytlin@moeacgs.gov.tw (Y.-T.L.); hjj@moeacgs.gov.tw (C.-C.H.)

* Correspondence: shihmeng@mail.ntou.edu.tw; Tel.: +886-2-2462-2192 (ext. 6171)

**Featured Application: Investigating the spatial and temporal dynamic behavior of groundwater level fluctuation in mountainous areas using the proposed approach can assist in obtaining effective strategies for developing and managing groundwater resources.**

**Abstract:** The analysis of the spatiotemporal characteristics of groundwater level variation is a prerequisite for evaluating groundwater potential or underpinning aquifer sustainability development in hydrogeological engineering practices. This study explores the dominant influencing factors that control groundwater dynamics and develops an estimation of groundwater level fluctuation (GWLF) potential in the complex aquifer systems of mountainous areas. Eight natural environmental factors, including slope, drainage density, land use, lithology, hydraulic conductivity, porosity, groundwater depth, and regolith thickness, have been selected as influencing factors, and the feature scores for different factors associated with GWLF potential were given with the expert scoring method. The weighting coefficients of individual influencing factors for wet/dry seasons were determined using the pseudo-inverse method based on the groundwater level data of 18 well stations observed from 1 November 2011 to 31 October 2019 in Taiwan mountainous areas. The results show that the weighting coefficients of these factors in controlling GWLF potential are variable and affected by seasonal and annual rainfall conditions. Based on the determined weighting coefficients, the spatial distribution of GWLF potential can be effectively produced. Finally, the simulated GWLF potential results were verified by comparing the observed data. The verification result demonstrates that the developed model can predict the spatial GWLF distribution based on the groundwater level data from a few 'wells.

**Keywords:** groundwater dynamics; influencing factors; groundwater level fluctuation potential; pseudo-inverse method; weighting coefficient

## 1. Introduction

Groundwater is necessary for human survival and development. The exploitation and conservation of groundwater resources are essential in underpinning sustainable aquifer development. In recent decades, extreme rainfall caused by climate change has impacted subsurface hydrologic systems [1,2]. As climate variability intensifies, groundwater resources may alter, and water shortages may become more pronounced. Due to the frequent drought in many countries worldwide, groundwater resources stored in fractured rock masses in mountainous areas have gradually become alternative water resources to combat

this natural disaster [3,4]. In addition, much of the groundwater, either recent anthropocene groundwater [5] or older fossil groundwater [6], stored under porous and fractured rocks is vulnerable to water quality deterioration and groundwater depletion at geographically varying locations. Therefore, assessing the groundwater resource potential and quality in mountainous areas is a prerequisite.

In Taiwan, groundwater resources are traditionally extracted from the plain area commonly composed of porous media. However, severe climate change and subsidence hazards in the plain areas of Taiwan have created restrictions on using such water resources [7]. The mountainous area of Taiwan, which occupies two-thirds of this island area, may offer the potential significance of freshwater compared to groundwater from the plain area of Taiwan. While determining whether aquifers in the mountain region can be considered an essential source of supplementary water resources, the first-ever catchment-scale Groundwater Investigation in the Mountainous region of Taiwan (GIMT) was initiated by the Central Geological Survey, Ministry of Economic Affairs (MOEA) of Taiwan in 2010 [8]. The main objectives of this project are to collect basin-scale hydrogeological data, build groundwater level observation wells for collecting continuous groundwater level changes, and develop a hydrogeological database for Taiwan. Although the project has pointed out the groundwater potentials of various formations from pumping tests performed in the existing boreholes, the groundwater level data continuously collected have not been fully utilized. One of the applications for such data can be to investigate groundwater's spatial and temporal dynamic behavior in mountainous areas, which may further help to determine effective strategies for developing and managing groundwater resources.

Long-term groundwater level data can reflect changes in flow and gradients over time and space under natural and artificial effects [9]. Groundwater levels may rise or fall depending on various influencing factors that can be site-specific. Meanwhile, the groundwater level fluctuation pattern is an excellent signal for investigating the potential factors that affect groundwater dynamic behaviors, a hydraulic connection between aquifers, and the interaction between the surface water and subsurface water [10]. Previous groundwater studies pointed out that the groundwater fluctuation potential depends on many environmental factors, such as the lithology and hydraulic characteristics of formations, geomorphology, topography, land use/land cover type, regolith thickness, drainage density, and evapotranspiration [11–20]. Rainfall and snowmelt are the primary external factors that affect seasonal and annual groundwater level changes [21–24]. In addition to rain under normal climate conditions, extreme rainfall caused by climate change may influence groundwater level changes. Allen et al. [25] observed the relationship between groundwater level changes and climate changes through observation wells, climate, and hydrological data. The average monthly groundwater level can be assessed as a seasonal cycle dominated by rainfall and snowmelt. The nature of climate change is different and complicated, showing the complexity of the hydrological conditions and the groundwater system. Additionally, most of the related studies used various influencing factors and an assumed weight for the corresponding factor to map the potential groundwater recharge areas or high groundwater yield zones. Related studies rarely apply groundwater level data to derive the weights of influencing factors. Whether the weighting coefficient of each factor altered with seasonal and annual groundwater level changes has not been addressed. Therefore, this study evaluates the groundwater resource potential in Taiwan's mountainous areas based on the seasonal and annual changes in groundwater levels.

In this study, the groundwater level data of 18 well stations monitored from November 2011 to October 2019 in the Taiwan mountainous region of the Choushui River Basin and Pei-Kang River Basin were used to evaluate and discuss variations in the groundwater level during the wet and dry seasons in different years. Possible influencing factors that affect groundwater level fluctuation potential were proposed and investigated. Furthermore, a technique for mapping the spatial groundwater level fluctuation potential was developed. Meanwhile, the weightage of each influential factor to control the degree of groundwater level fluctuation was determined. Weighting coefficients of influencing factors

in the wet/dry seasons for different hydrological years were also discussed. Finally, the groundwater level monitoring data were utilized to verify the estimated groundwater level fluctuation potential.

## 2. Study Area

The study area has a central geographic location of 23°43′ N and 121°56′ E, with a total coverage of 3084 km$^2$, mainly in Central Taiwan. The elevations range from 100 m to 3927 m. The primary investigation river basin includes the Pei-Kang River Basin and the Chou-Shui River Basin. The geological zone covers from the west to the east, including the coastal plain, the western piedmont, and the Hsuehshan mountain. The groundwater level observation wells were built in different geological units, as shown in Figure 1. According to the rock core data from borehole drilling, the lithologies in the study area are diverse. The predominant lithologies include conglomerate, mudstone with conglomerate, sandstone, shale, alternations of sandstone and shale, siltstone, argillaceous siltstone, mudstone, quartzite, phyllite, and slate. The groundwater level data from November 2011 to October 2019 selected from 18 groundwater monitoring sites in the study area were utilized to determine the spatial and temporal characteristics of the groundwater level change. In addition, the annual rainfall ranges from 1205 mm to 5166 mm, and the average annual rainfall is 2862 mm.

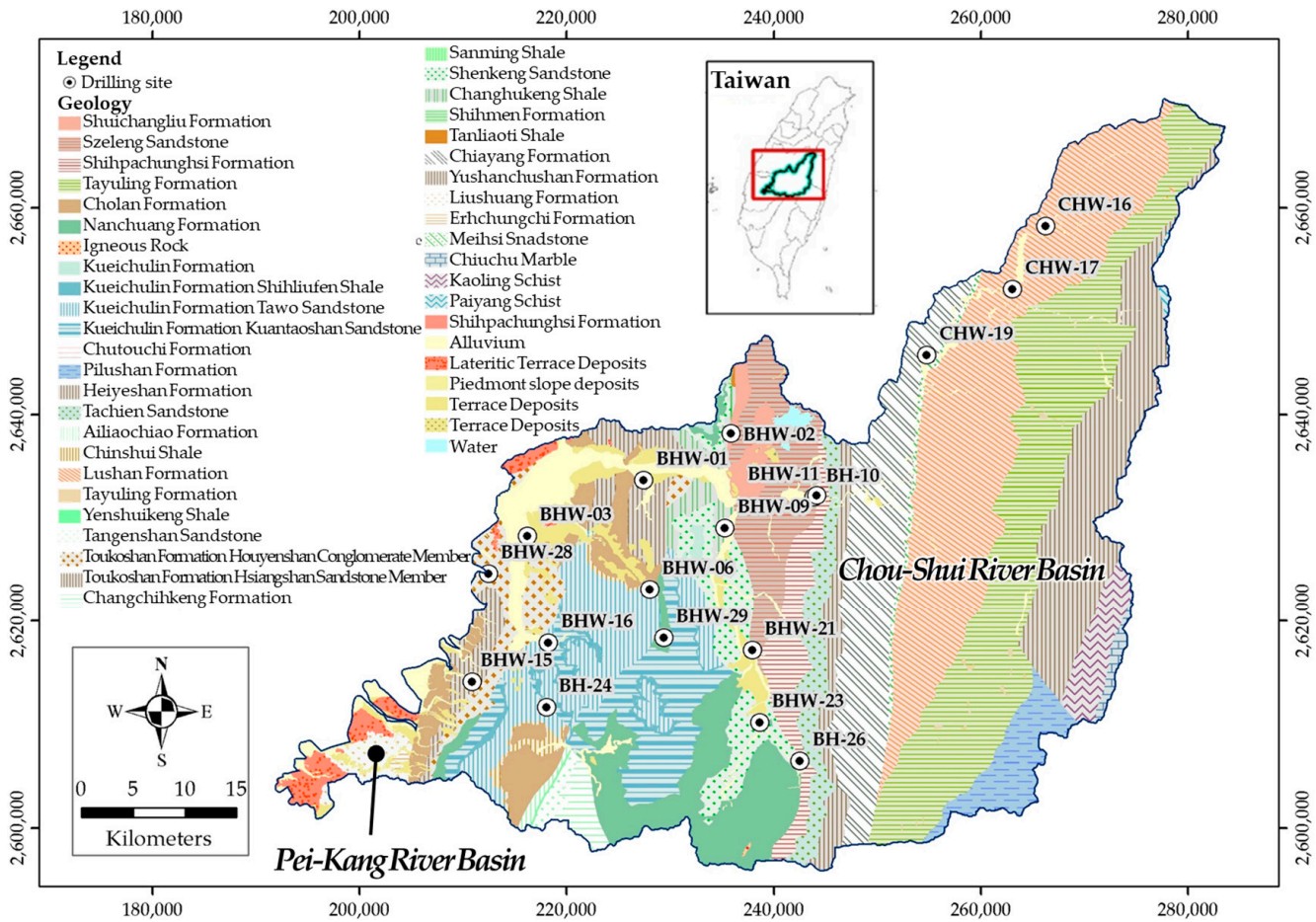

**Figure 1.** Location map of the study area and the 40 groundwater monitoring wells.

Each groundwater monitoring site is designed as a two-level groundwater monitoring system, as depicted in Figure 2. The shallow groundwater levels reflect the groundwater dynamics at the regolith zone and the highly fractured bedrock. The deeper zone is to observe the change in the groundwater level in the fractured rock mass. From the groundwater resource development in Taiwan's mountainous areas, discovering the variations in

the groundwater levels in the shallow zone is more significant than those in the deep zone. Therefore, variations in the shallow groundwater level are the main objectives of this study.

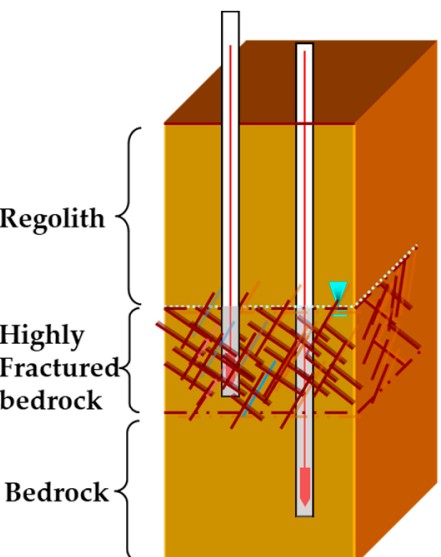

**Figure 2.** Schematic diagram of a two-level groundwater monitoring system.

## 3. Methodology

The external factors that stimulate groundwater level changes can be classified into two categories. The first one is triggered by rainfall conditions. The other is activated by stress changes in aquifers, such as atmospheric pressure, tide, or earthquakes [26–28]. Generally speaking, the second category is supposed to have more minor variations in the groundwater level than the first category. This study focused on the first category, which has more of an influence on groundwater level fluctuations. If the groundwater system is stimulated by rainfall, some possible factors may affect the following change in aquifer storage, including the topography, geomorphology, lithology, groundwater depth, and hydrogeological characteristics of the aquifer. Therefore, possible influencing factors associated with rainfall were adopted to establish the static potential for groundwater level change and to classify the potential for groundwater level change according to the influence of each factor. In addition, the feedback from the actual groundwater level monitoring data was used to investigate the dynamic behavior of groundwater in the mountain area and to understand the causes and types of groundwater level changes in the mountain area. The results can be used to provide the future construction of groundwater flow and groundwater recharge analysis in mountainous areas. The related theoretical approaches are described below.

### 3.1. Selected Influencing Factors

The influencing factors considered in this study include elevation, drainage density, slope, land use, lithology, depth to the water table, regolith thickness, porosity, and hydraulic conductivity. Overall, the selected factors can be attributed as geomorphologic, geological, and hydrogeological factors. The influencing factors are described as follows.

### 3.1.1. Slope

The slope is calculated by using a 20 m × 20 m DTM (Digital Terran Model) provided by the Ministry of the Interior, Taiwan. As shown in Figure 3, the slope distribution in the upstream is steeper. Overall, the slope gradually becomes gentle from the upstream to the downstream. The slope has a direct relationship with groundwater level fluctuation. Groundwater recharge comes from rainfall, and then the slope directly affects runoff and infiltration. Usually, rain does not have enough time to infiltrate into the subsurface at the steep slope during rainfall periods.

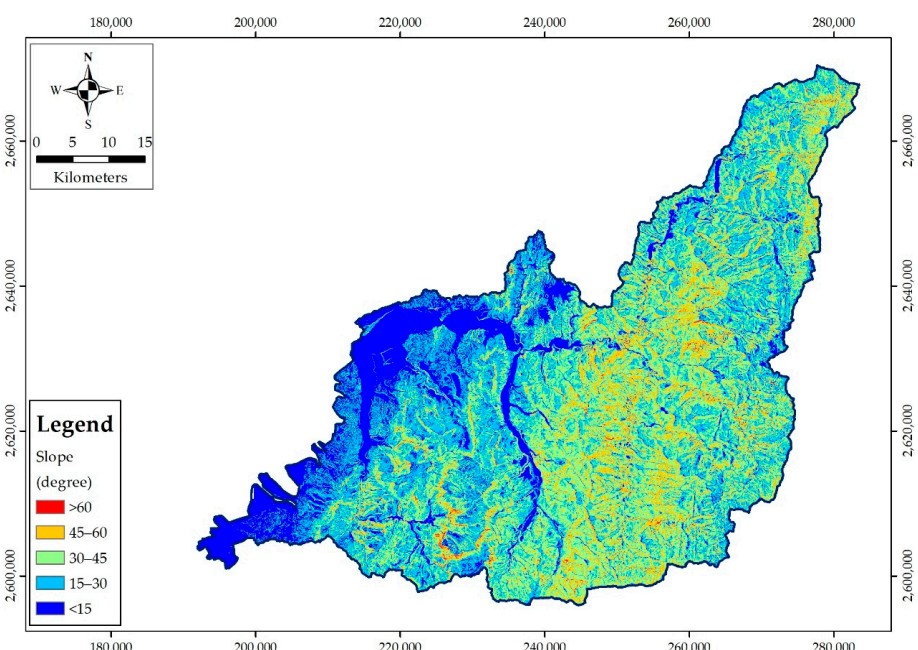

**Figure 3.** Slope distribution of the study area.

### 3.1.2. Drainage Density

Drainage density is the ratio of the length of all stream orders to the watershed area. Since the river represents the location of the surface water body, there is a significant correlation between the drainage density and the change in the groundwater level. The higher the drainage density, the greater the groundwater level fluctuation. The drainage density was also determined based on a 20 m × 20 m DTM using the ArcGIS tool.

$$DD = LL/A \tag{1}$$

where DD is the drainage density and its unit is the reciprocal of the length; A is the watershed area; and LL is the total stream length of all stream orders. As shown in Figure 4, the drainage density of the study area was classified into four classes.

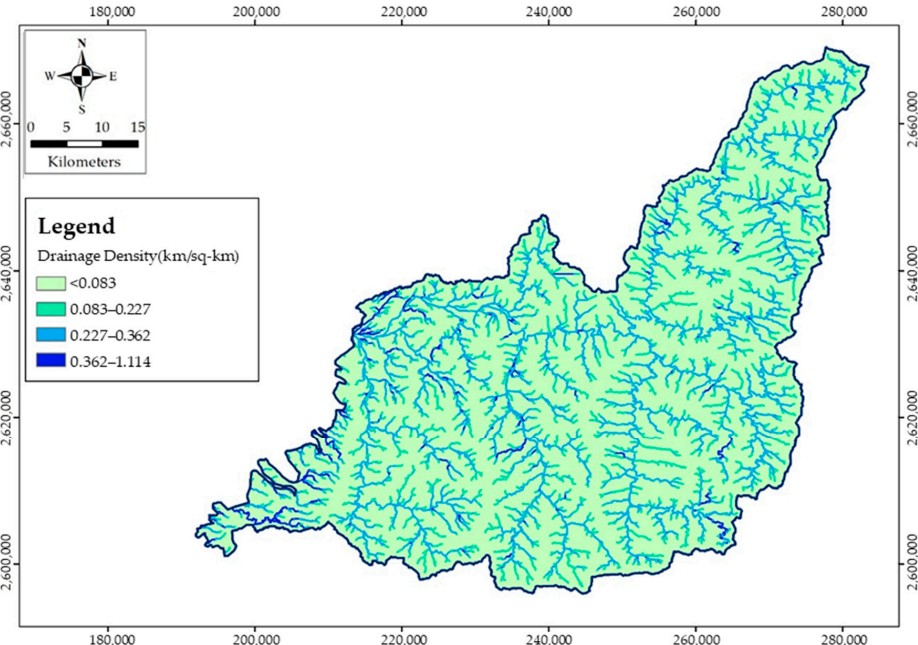

**Figure 4.** Drainage density distribution of the study area.

### 3.1.3. Land Use

The existing land use map produced by the survey and assessment of groundwater resources in the middle mountainous region of Taiwan was used [8]. The grid resolution of the map is 500 m × 500 m. In this study, the grid resolution is unified to resample to 40 m × 40 m using the ArcGIS spatial analysis module. As shown in Figure 5, the land use type can be classified as the built-up land, forest, exposed area, crop land, and water bodies.

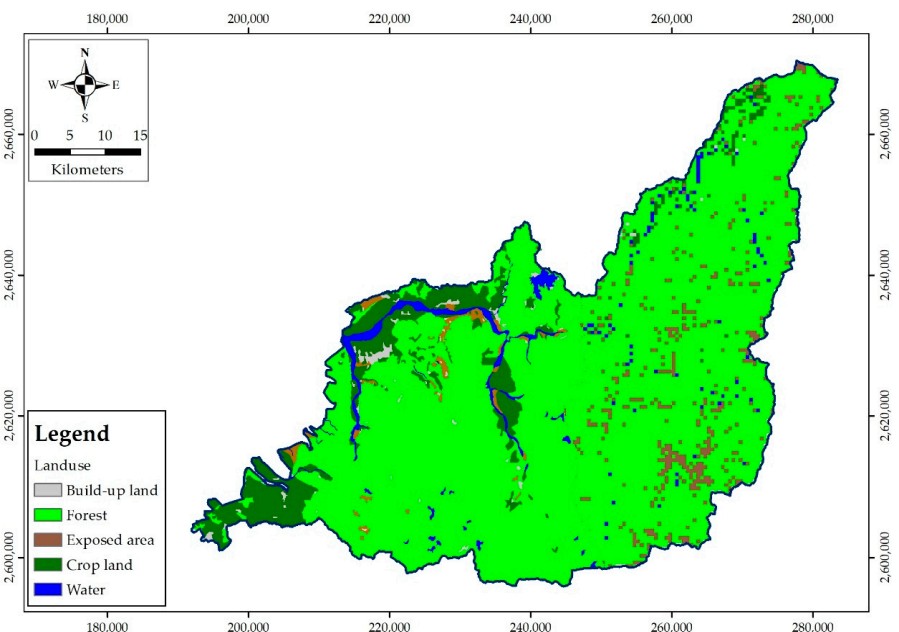

**Figure 5.** Land use distribution of the study area.

### 3.1.4. Lithology

Lithology has a significant influence on water infiltration [29]. Thus, based on the official geological map issued by the Taiwan Central Geological Survey [8], the lithological factor can be classified as shale, sandstone & shale (mainly shale), sandstone & shale (mainly sandstone), quartzite, slate & phyllite, and unconsolidated rock, as shown in Figure 6.

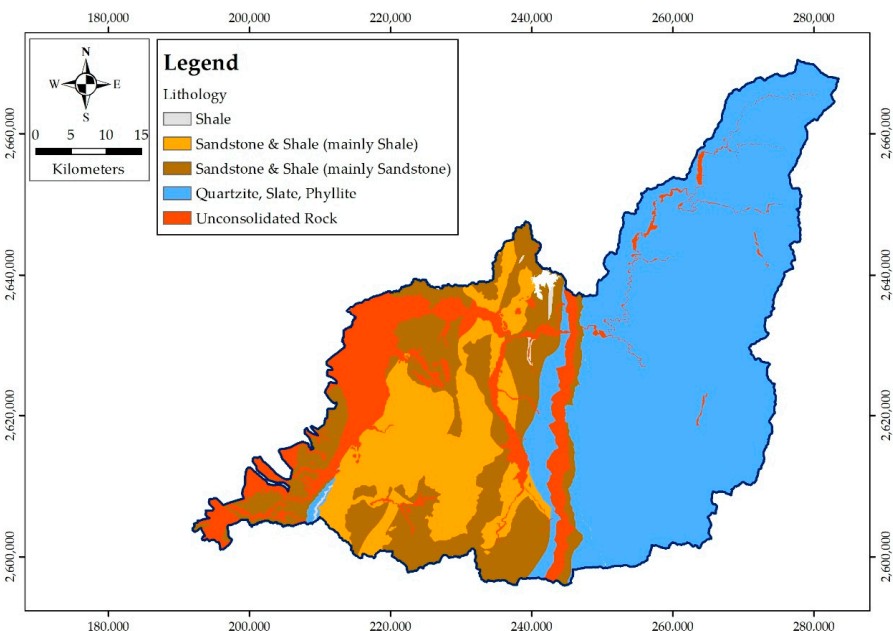

**Figure 6.** Lithology distribution of the study area.

### 3.1.5. Hydraulic Conductivity

Hydraulic conductivity is an indicator for evaluating the ability of the formation to transmit groundwater through pore spaces and fractures. If fissures or cracks prevail in a fractured rock aquifer, the aquifer appears to be more permeable, and groundwater in the highly fractured rock aquifers can move faster. The greater the hydraulic conductivity of the formation, The greater the degree of variation in the groundwater level.

This study obtained the hydraulic conductivity values along each well from in situ hydraulic tests. By using the above hydraulic conductivity data and the existing geological map, the spatial distribution of hydraulic conductivity can be produced, as shown in Figure 7. In this study, hydraulic conductivity is classified into five classes. In addition to the highest and lowest classes, each class is designed based on the difference of one order of magnitude to distinguish the formation's hydraulic property.

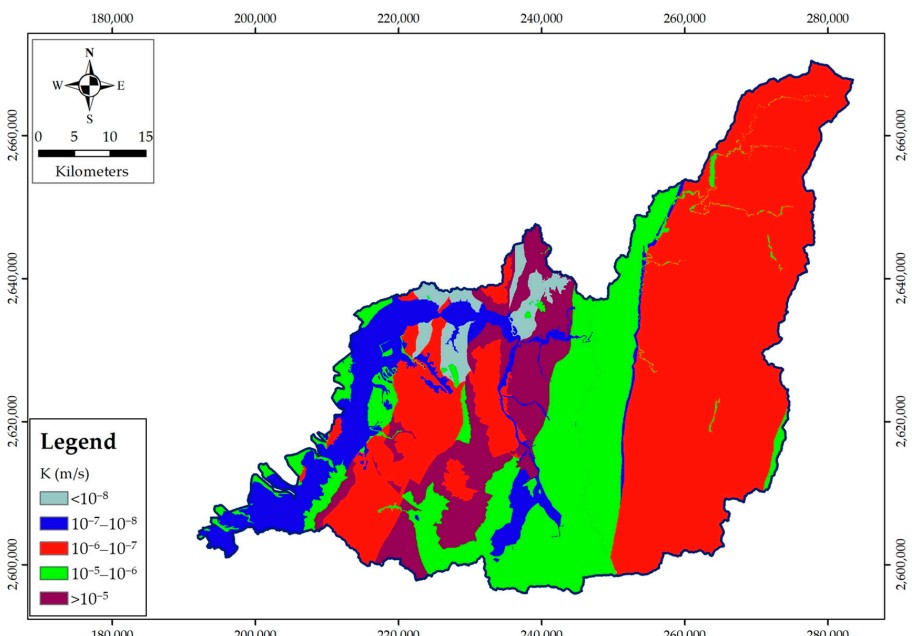

**Figure 7.** Hydraulic conductivity distribution of the study area.

### 3.1.6. Porosity

The porosity can be influenced by the physical properties of rock mass, such as hydraulic conductivity. The Kozeny–Carman equation theoretically expresses the relation between porosity and hydraulic conductivity [30]. According to the concept, this study selected this parameter as an influencing factor in studying GWLF potential. The porosity data were obtained from in situ sonic logging from the 18 boreholes. The value of porosity ranges from 1.32% to 30%. The field test data integrated with the existing hydrogeological map [8] were used to estimate the spatial distribution of porosity, as shown in Figure 8. By comparing the geological map, the porosity values associated with the type of lithology were also found.

### 3.1.7. Depth to the Water Table

Previous studies [8] have shown that the depth to the water table in a given well is related to the surface topography to some extent. In a flat area, the water table is closer to the ground; in a steeper area, the water table is farther from the ground. Thus, the depth to the water table may correlate with the ground elevation in a given well.

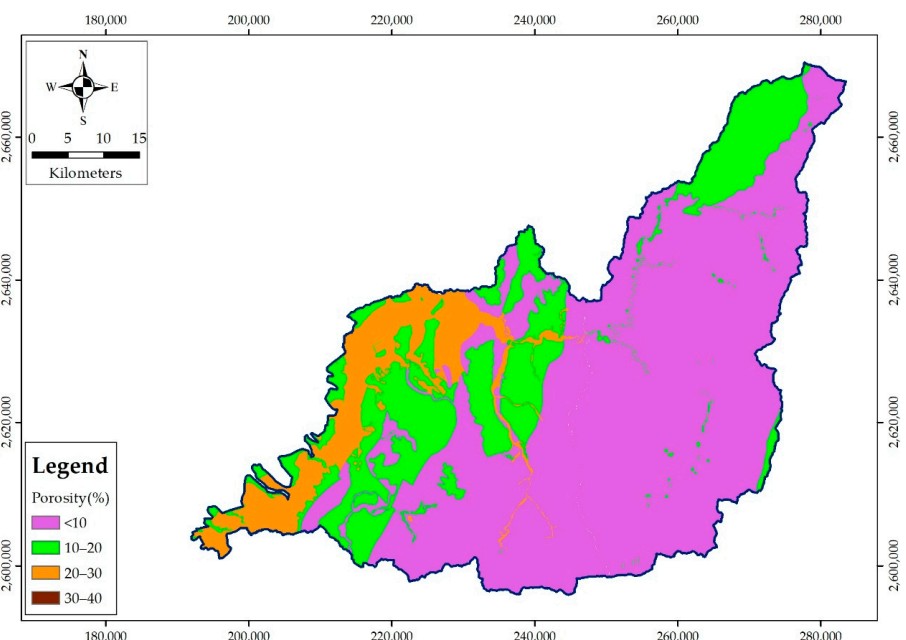

**Figure 8.** Porosity distribution of the study area.

In this study, 18 pairs of the depth to the water table data and the well elevation data were collected. A regression analysis was performed to establish a relationship between two variables. The regression analysis result is given in the following.

$$\text{DWT} = 21.22 + 0.0018(\text{GE}) \left(R^2 = 0.996\right) \tag{2}$$

where DWT is the depth to the water table; GE is the ground elevation of a given well. Based on the results of the R-squared values, the developed relationship between the depth to the water table and the ground elevation is very strong. Equation (2) can be used to generate the spatial distributions of the depth to the water table for the study area. Figure 9 shows the spatial distribution of the depth to the water table for the entire study area. The depth to the water table is smaller in areas close to rivers and larger in mountainous terrain.

3.1.8. Regolith Thickness

In general, regolith in the mountainous area of Taiwan consists of colluvium deposits with a large pore space, which may have great groundwater storage potential and affect changes in groundwater levels. Thus, the groundwater storage potential may positively vary with the regolith thickness (RT). Thus, the factor of the regolith thicknesses was chosen for the GWLF potential analysis.

To construct the spatial distribution of the regolith thickness of the study area from limited survey data, a regression analysis was performed to estimate the regolith thickness (RT) based on two factors: slope (S) and groundwater elevation (GE), which are also suggested by Hsu et al. [7]. The data required for the regression analysis were obtained from the survey results of 18 borehole logs in the study area. The following equation provides the regression model with a coefficient of determination of 0.75. The spatial distribution of the regolith thickness factor is illustrated in Figure 10.

$$\text{RT} = \frac{382}{\left(1 - \frac{\text{GE} - 387}{329}\right)^2 \times \left(1 - \frac{S + 18}{5}\right)^2} \tag{3}$$

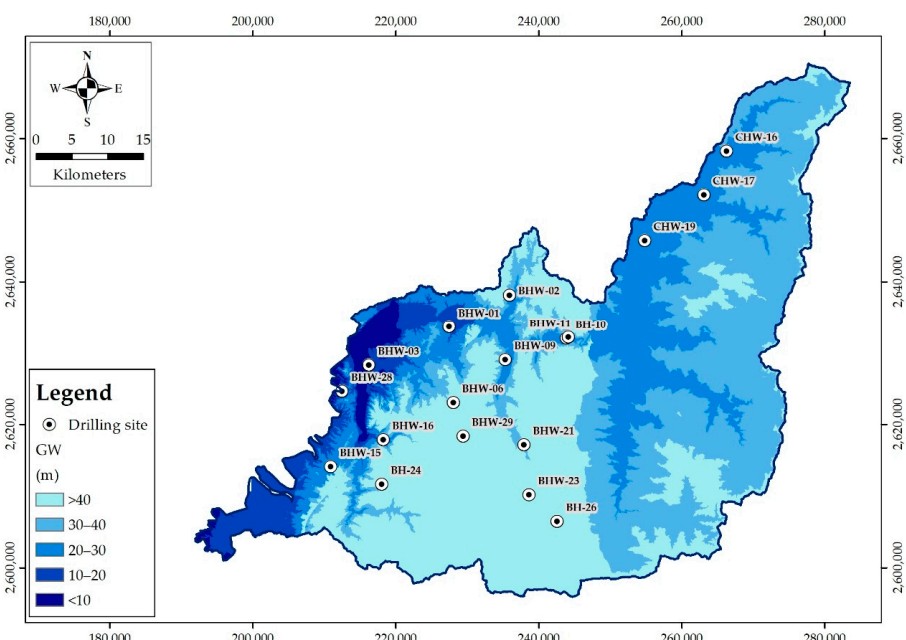

**Figure 9.** Depth to the water table distribution of the study area.

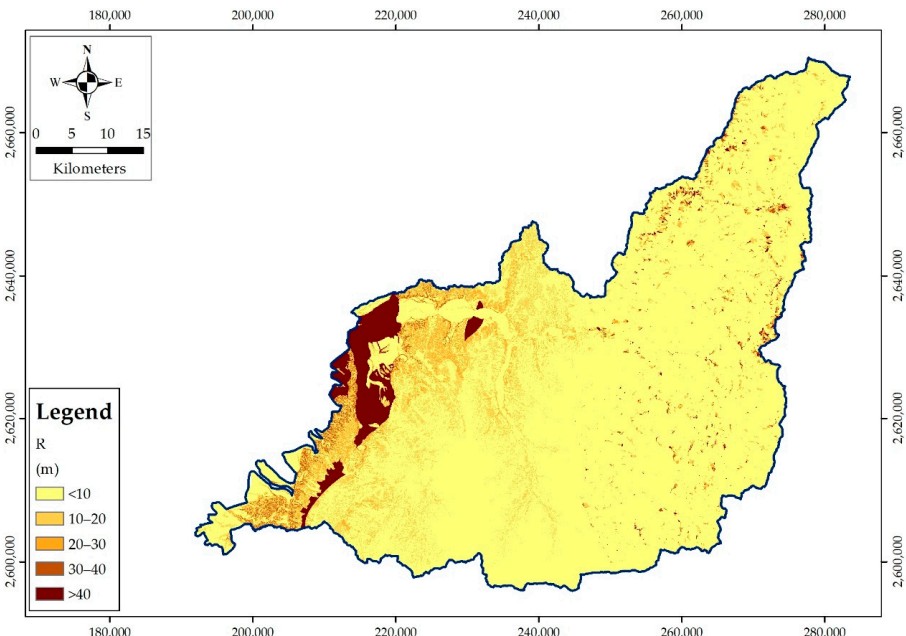

**Figure 10.** Regolith thickness distribution of the study area.

### 3.2. Designs of Feature Scores of Individual Influencing Factor Layers

Feature scores for different influencing factor layers with the expert assessment method were prioritized and assigned according to their influences on the GWLF potential, as shown in Table 1. The higher the feature score in each influencing factor, the more significant the influence on the groundwater level fluctuation. By using the ArcGIS spatial analysis module and the extracted feature score data from 18 drilling boreholes, the spatial distribution of the feature score for each influencing factor is illustrated in Figure 11.

**Table 1.** Assigned feature scores for different influencing factors associated with groundwater level fluctuation potential.

| Category | Influencing Factor | Upper Bound | Lower Bound | Feature Score | GWLF Potential |
|---|---|---|---|---|---|
| Geomorphologic Factor | Slope [S] (degree) | 90 | 60 | 1 | Low |
| | | 60 | 45 | 2 | |
| | | 45 | 30 | 3 | ↓ |
| | | 30 | 15 | 4 | |
| | | 15 | 0 | 5 | High |
| | Drainage density [D] (km/km²) | 0.083 | 0 | 1 | Low |
| | | 0.227 | 0.083 | 2 | |
| | | 0.362 | 0.227 | 3 | ↓ |
| | | 1.114 | 0.362 | 4 | High |
| | Land use [LU] | Built-up land | | 1 | Low |
| | | Forest | | 2 | |
| | | Exposed area | | 3 | ↓ |
| | | Crop land | | 4 | |
| | | Water | | 5 | High |
| Geological Factor | Lithology [LT] | Shale | | 1 | Low |
| | | Sandstone & shale (mainly shale) | | 2 | |
| | | Sandstone & shale (mainly sandstone) | | 3 | ↓ |
| | | Slate, Phyllite, Quartzite | | 4 | |
| | | Unconsolidated Rock | | 5 | High |
| | Regolith thickness [RT] (m) | < | 10 | 1 | Low |
| | | 10 | 20 | 2 | |
| | | 20 | 30 | 3 | ↓ |
| | | 30 | 40 | 4 | |
| | | > | 40 | 5 | High |
| Hydrogeological Factor | Hydraulic conductivity [K] (×10⁻⁹ m/s) | 10 | 0 | 1 | Low |
| | | 100 | 10 | 2 | |
| | | 1000 | 100 | 3 | ↓ |
| | | 10,000 | 1000 | 4 | |
| | | > | 10,000 | 5 | High |
| | Porosity [P] (%) | 10 | 0 | 1 | Low |
| | | 20 | 10 | 2 | |
| | | 30 | 20 | 3 | ↓ |
| | | 40 | 30 | 4 | High |
| | Depth to the water table [GW] (m) | > | 40 | 1 | Low |
| | | 30 | 40 | 2 | |
| | | 20 | 30 | 3 | ↓ |
| | | 10 | 20 | 4 | |
| | | < | 10 | 5 | High |

↓: The direction of GWLF potential from low to high.

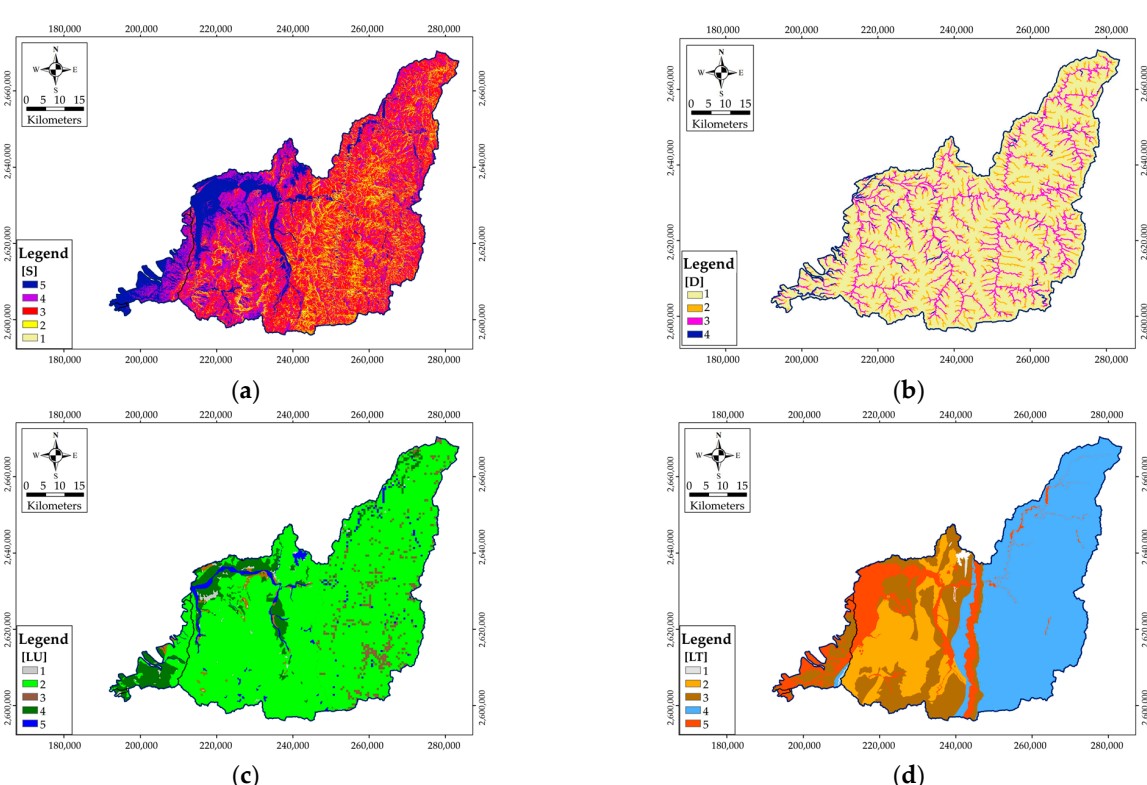

**Figure 11.** *Cont.*

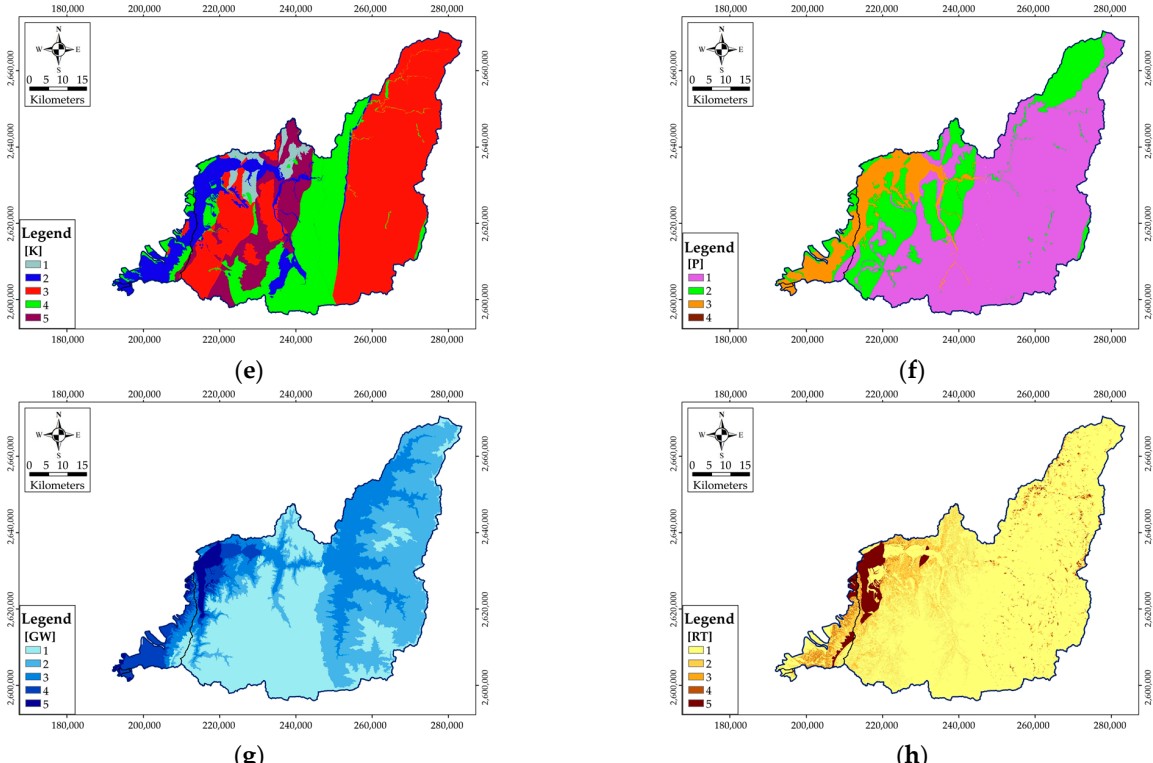

**Figure 11.** The spatial distributions of the feature score for different influencing factors: (**a**) slope [S], (**b**) drainage density [D], (**c**) land use [LU], (**d**) lithology [LT], (**e**) hydraulic conductivity [K], (**f**) porosity [P], (**g**) depth to the water table [GW], (**h**) regolith thickness [RT].

*3.3. Determination of the Weightings of Individual Influencing Factors from Groundwater Level Fluctuation Data*

As stated in Section 3.1, the groundwater level fluctuation may be affected by the selected eight influential factors. Each influential factor has its weightage to control the degree of groundwater level fluctuation. While determining the weights of individual influencing factors from the feedback of groundwater level fluctuation, the following equation can be used, as shown in Figure 12.

$$[P]_{m \times n} \times [W]_{n \times 1} = [\Delta h]_{m \times 1} \tag{4}$$

where [P] is the feature scores of influential factors from the groundwater monitoring wells, [W] is the estimated weighting coefficients for eight influential factors, and [Δh] is a specific value of groundwater level fluctuation for each groundwater monitoring well in a period. Due to the [P] matrix, which is a non-symmetric matrix in Equation (4), the weighting coefficients are estimated using the Pseudo-inverse method [31]. The weighting coefficient can describe the relative influence of each factor on the groundwater fluctuation potential.

This study used the ArcGIS toolbox to construct the thematic map for each selected natural environmental factor, as shown in Figure 11. By using the spatial data in these layers, the feature scores corresponding to the location of each station well were obtained. Since 18 groundwater monitoring wells are located in the study area, an 18 × 8 matrix [P] can be constructed. As for the matrix [Δh], it can be determined from the groundwater monitoring data of 18 wells. The data of groundwater level fluctuations are based on the average fluctuating water levels during the wet and dry periods from November 2011 to October 2019, in which the wet season is from May to October and the dry season is from November until April of the following year, respectively. Table 2 shows the maximum variation in the groundwater level of each borehole during the wet and dry seasons. The results show that, during the wet season for the Year 2011 to 2014, the groundwater level fluctuation of BHW-09 is the largest, followed by BHW-29 and BHW-06, and BHW-26 is

the smallest. However, the above conclusion for ranking the groundwater level fluctuation potential in groundwater level observation wells does not apply to other observation years. For the dry season, the ranking of the potential for groundwater level changes in the groundwater level observation wells from eight years of data shows no particular pattern. Regarding this lack of certain regularity, this research speculates that it should be related to Taiwan's uneven spatial distribution of rainfall. However, this speculation still needs further analysis and confirmation, but it is not within the scope of this research.

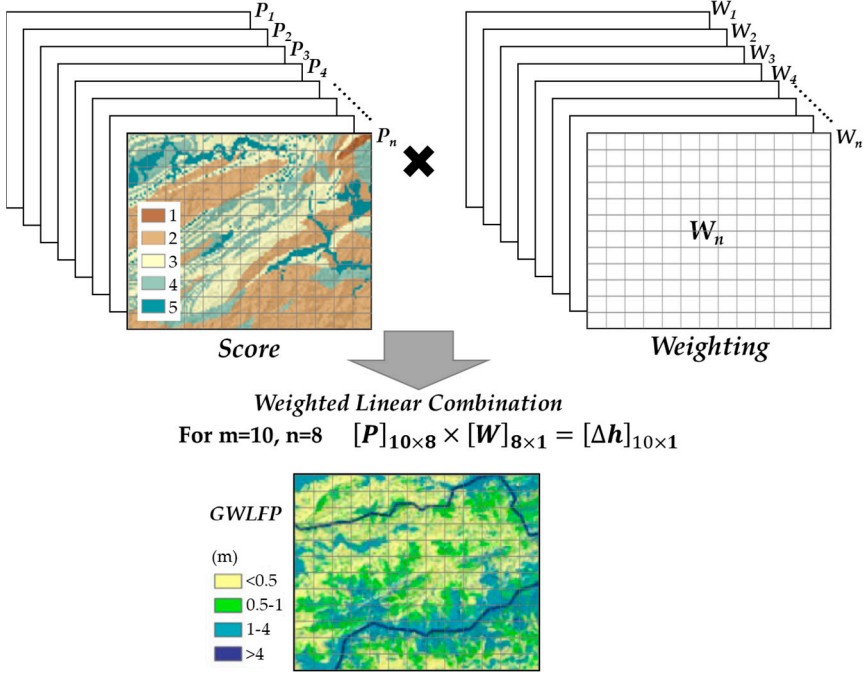

**Figure 12.** The schematic diagram for the analysis of groundwater level fluctuation potential.

**Table 2.** Statistics of variation of the groundwater level in each monitoring well.

| Well No | November 2011–October 2012 D | W | November 2012–October 2013 D | W | November 2013–October 2014 D | W | November 2014–October 2015 D | W | November 2015–October 2016 D | W | November 2016–October 2017 D | W | November 2017–October 2018 D | W | November 2018–October 2019 D | W |
|---|---|---|---|---|---|---|---|---|---|---|---|---|---|---|---|---|
| BHW-01 | 2.04 | 3.89 | 1.28 | 2.20 | 0.51 | 1.57 | 0.45 | 1.68 | 0.99 | 1.92 | 1.55 | - | 2.25 | 2.35 | 0.79 | 2.22 |
| BHW-02 | 3.47 | 4.37 | 3.45 | 4.43 | 1.25 | 4.59 | 2.17 | 4.99 | 2.56 | 3.52 | 3.42 | - | 4.13 | 3.95 | 3.71 | 3.00 |
| BHW-03 | 1.10 | 2.94 | 1.39 | 2.49 | 0.40 | 1.95 | 0.93 | 2.32 | 0.72 | 1.53 | 0.66 | 2.31 | 0.55 | 2.17 | 0.72 | 2.16 |
| BHW-06 | 5.16 | 8.66 | 7.17 | 8.18 | 1.76 | 8.44 | 1.86 | 8.81 | 6.32 | 7.44 | 3.17 | 9.09 | 4.12 | 7.00 | 3.75 | 7.70 |
| BHW-09 | 2.32 | 15.46 | 3.90 | 16.14 | 1.20 | 14.16 | 1.27 | 7.06 | 4.38 | 6.72 | 5.55 | 17.98 | 5.85 | 18.67 | 2.56 | 15.47 |
| BH-10 | 4.27 | 4.97 | 6.31 | 4.74 | 2.09 | - | 5.23 | 5.74 | 6.35 | 6.61 | 5.14 | 6.78 | 1.95 | 3.56 | 3.44 | 4.71 |
| BHW-11 | 1.72 | 4.35 | 2.73 | 3.97 | 1.36 | 3.58 | 1.25 | 3.45 | 2.58 | 3.05 | 2.29 | 7.03 | 1.39 | 1.82 | 1.39 | 4.89 |
| BHW-15 | 2.08 | 2.98 | 5.75 | 3.65 | 2.22 | 2.60 | 0.69 | 2.64 | 3.82 | 2.75 | 0.82 | 2.00 | 1.68 | 3.48 | 1.61 | 2.89 |
| BHW-16 | 1.17 | 8.37 | 0.96 | 6.67 | 0.70 | 6.40 | 0.62 | 5.36 | 1.19 | 5.19 | 1.12 | 8.44 | 1.20 | 6.66 | 0.45 | 6.19 |
| BHW-21 | 1.02 | 8.19 | 1.76 | 6.49 | 0.64 | 3.06 | 0.69 | 2.40 | 1.49 | 3.20 | 0.99 | 9.43 | 1.04 | 4.24 | 1.45 | 6.37 |
| BHW-23 | 0.93 | 6.98 | 1.35 | 4.16 | 1.03 | 3.32 | 0.88 | 2.62 | 1.54 | 2.23 | 1.28 | 7.86 | 0.85 | 2.76 | 0.85 | 5.29 |
| BH-24 | 2.20 | 4.23 | 1.93 | 3.55 | 0.90 | 3.58 | 1.14 | 5.29 | 5.37 | - | - | 5.75 | 2.22 | 2.50 | 1.16 | 2.90 |
| BH-26 | 1.03 | 2.00 | 0.77 | 1.39 | 0.89 | 1.46 | 0.87 | 1.47 | 0.57 | 1.29 | 0.63 | 2.59 | 0.74 | 1.30 | 0.45 | 1.70 |
| BHW-28 | 2.03 | 7.48 | 0.57 | 6.66 | 0.77 | 6.32 | 0.90 | 5.41 | 1.97 | 6.03 | 0.87 | 6.41 | 0.66 | 7.44 | 0.86 | 8.14 |
| BHW-29 | 2.24 | 10.65 | 1.91 | 9.79 | 2.64 | 10.57 | 2.74 | 9.36 | 2.22 | 12.17 | 1.75 | 8.88 | 1.99 | 10.20 | 4.49 | 14.46 |
| CHW-16 | 3.13 | 8.09 | 5.82 | 6.53 | 3.33 | 5.93 | 2.93 | 5.72 | 5.02 | 3.19 | 2.86 | 9.06 | 1.67 | 3.44 | 1.75 | 7.51 |
| CHW-17 | 4.19 | 6.23 | 4.70 | 5.88 | 2.02 | 5.77 | 2.69 | 5.49 | 5.00 | 5.04 | 6.00 | 7.82 | 5.16 | 6.17 | 4.03 | 6.33 |
| CHW-19 | 4.41 | 5.06 | 4.03 | 3.83 | 0.99 | 4.51 | 1.19 | 4.35 | 3.62 | 3.51 | 3.92 | 4.64 | 0.85 | 3.85 | 4.07 | 4.31 |

D: Dry season from November to April of next year. W: Wet season from May to October of each year. -: Data were not available due to the repair of the water pressure gauge. Unit: m.

## 4. Results and Discussion

### 4.1. Weighting Coefficients of Influencing Factors in the Wet/Dry Season for Different Hydrological Years

The [Δh] matrix in Equation (4) can be composed of groundwater level fluctuation data in space. The GWLF data from the available groundwater monitoring wells in a period can be used to construct the [Δh] matrix. By using the [Δh] matrix and the given [P] matrix, as shown in Figure 12, the weighting coefficients of the influencing factors

through Equation (4) can be obtained. In this way, the abilities of various factors to control groundwater level fluctuations in a period can be investigated.

Figure 13 shows the estimated weighting coefficients for eight influencing factors in different years' wet and dry seasons. Regardless of the season or year, the composition of weighting coefficients is quite different. The weighting coefficients among the eight factors for most years are relatively different in the wet season. The factors that have a more significant influence on GWLF include hydraulic conductivity (K), depth to the water table (GW), lithology (LT), drainage density (D), and regolith thickness (RT). In the dry season, the weighting coefficients among the eight factors for most years are relatively indistinguishable. Lithology (LT), slope (S), and porosity (P) are the factors that contribute more to the potential for groundwater level changes compared to other factors. In addition, the weighting coefficients of each factor in the dry season and wet season of the same year are also different. The primary factors affected by the season include hydraulic conductivity (K), depth to the water table (GW), and drainage density (D).

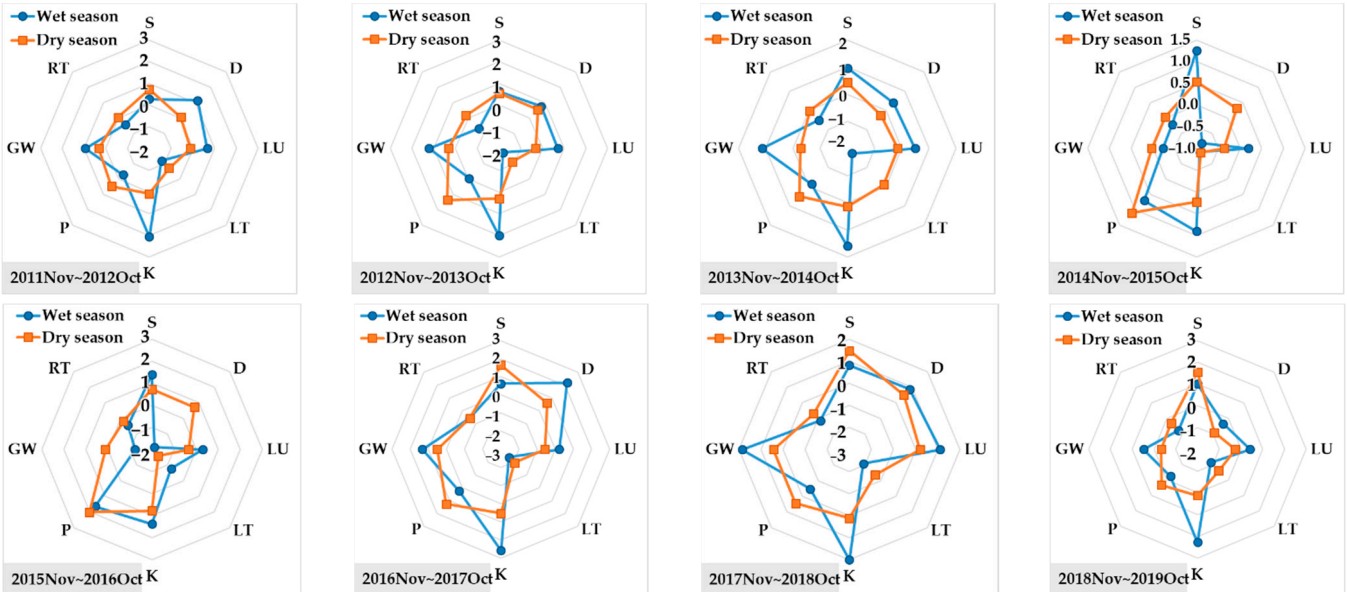

**Figure 13.** The estimated weighting coefficients in the wet and dry seasons for different years.

The above results show that the weighting coefficients of the factors are variable. Unlike the previous studies [9–18], the weighting coefficients for the selected factors may keep constants. By inspecting the possible reasons for the difference, the weighting coefficients for the selected factors may vary depending on rainfall patterns or groundwater recharge conditions. For example, there is a big difference between the accumulated rainfall in the wet and dry seasons in Taiwan; thus, rainfall replenishment into the formations may have a significant discrepancy. Furthermore, the annual accumulated rainfall in Taiwan is diverse due to climate change; thus, rainfall replenishment into the formations may also vary. Therefore, in this study, the weighting contribution of the influencing factors was obtained through the actual groundwater level change data, and then the calculated weighting coefficients were used to estimate the spatial groundwater level potential change. This newly proposed method can better capture the actual dynamic behavior of groundwater.

### 4.2. Comparision of Observed and Simulated GWLF Potential

To ensure the accuracy of the estimated GWLF results, the simulated results can be compared with the observed GWLF from 18 groundwater monitoring wells with 8 hydrological years of data. A scatter plot with a 45-degree reference line was used to verify the difference between the simulated and observed GWLF values. As shown in Figure 14, the value for the observed GWLF does tend to increase as the value of the simulated GWLF

increases for both types of seasons. The data points are randomly scattered around the reference line. This outcome indicates that this presented model is an unbiased model. In addition, the data points for the dry season are confined at approximately six meters of groundwater level fluctuation. The data points for the wet season are spread up to approximately 18 m of groundwater level fluctuation.

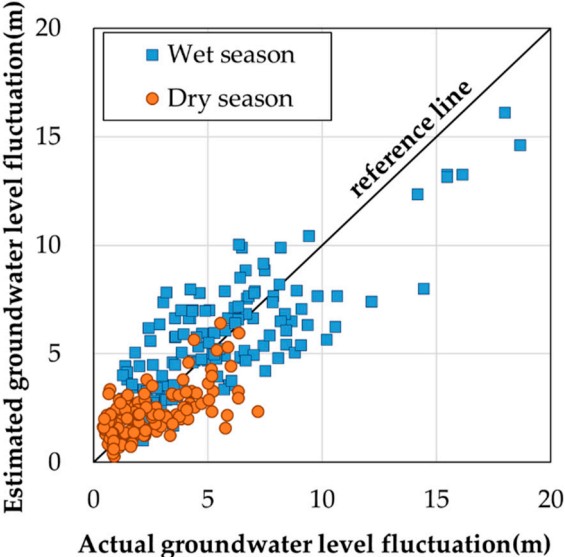

**Figure 14.** Comparison between the observed and estimated groundwater level fluctuation in the wet and dry seasons.

In addition to the visual interpretation, Equation (5) can be used to compute the percentage error between the estimated and observed GWLF values for each data point. The average error between the estimated and observed GWLF values from 144 measurements for the dry season is approximately 33.2%. The average error between the estimated and observed GWLF values from 144 measurements for the wet season is approximately 19.2%. Based on Donigian's [32] guidelines of error for calibration/verification tolerances to watershed model users, the percentage mean error value is within 20%, which indicates that the verified model is considered "very good"; the percentage mean error value is between 20% and 30%, which indicates that the verified model is considered "good"; the percentage mean error value is between 30% and 45%, which indicates that the verified model is considered "fair"; the percentage mean error value is over 45%, which indicates that the verified model is considered "poor". The above guidance is provided for the verification of sediment simulation in a watershed model. Although this guideline does not directly address the evaluation of GWLF simulation results, both GWLF and sediment parameters are considered challenging while performing simulations of hydrological models. Therefore, in the absence of a reference for the simulation of GWLF, this study refers to this guideline to evaluate the performance of the proposed model in simulating GWLF. Based on the guidance, the proposed model for predicting GWLF in the dry season can be considered "fair", but that in the wet season can be considered "good". Overall, the verification data demonstrate that the developed model can be expected to predict the GWLF for different seasons with a "good" level of accuracy.

$$\% \text{ error} = \left| \frac{\text{GWLF}_{estimated} - \text{GWLF}_{observed}}{\text{GWLF}_{observed}} \right| \times 100 \tag{5}$$

### 4.3. Spatial Distribution of GWLF Potential

After determining the weighting coefficients for different rainfall conditions, the spatial distribution of GWLF potential can be calculated from the feature score matrix [P] and estimated weighting coefficients in Figure 13. As shown in Figure 15, the groundwater

level fluctuation potential in the wet season is higher than that in the dry season. The groundwater level fluctuation potentials in different hydrological years are quite different. In particular, the potential for groundwater level change in the wet season is more pronounced than that in the dry season year. Regarding spatial outcomes, the potential of groundwater level fluctuation in the downstream of the study area is more significant than that in the upstream of the study area, regardless of the season type. The higher GWLF potential in the downstream area, in addition to the characteristics of influencing factors, may be due to lateral replenishment from the upstream area. In conclusion, by viewing the GWLF potential map, the spatial and temporal dynamic behavior of groundwater in mountainous areas can be clearly understood.

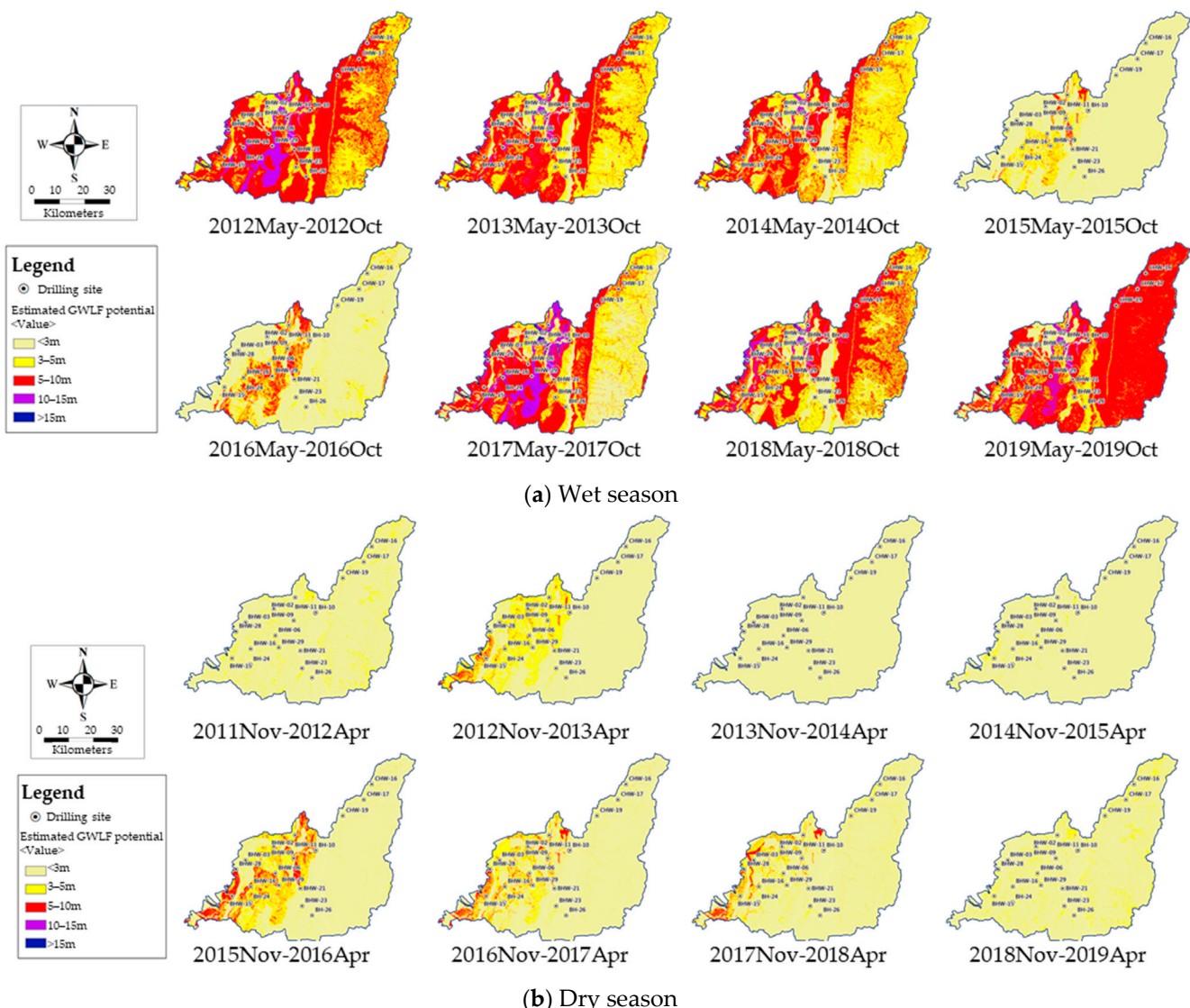

**Figure 15.** Estimated groundwater level fluctuation potential during 2011–2019 in the wet/dry season.

### 4.4. Relationship between Rainfall and Groundwater Level Changes

To investigate the relationship between rainfall and groundwater level changes, data from 2011 to 2019 were collected for analysis. The accumulated rainfall data in Figures 15 and 16a,b show the spatial distribution of accumulated rainfall and the estimated GWLF potential in the wet and dry seasons, respectively. As shown in both figures, the amount of rainfall is proportional to the estimated GWLF potential. By comparing the spatial distribution of rainfall with the GWLF data, a good correlation between the groundwater level change and seasonal rainfall for most of the wells is also found. The

groundwater level usually changes with the occurrence of rainfall. It can be inferred that the groundwater circulation is apparent. However, while rainfall stops, the groundwater level will return to its normal groundwater level, indicating that the aquifer cannot store groundwater easily in the mountainous area.

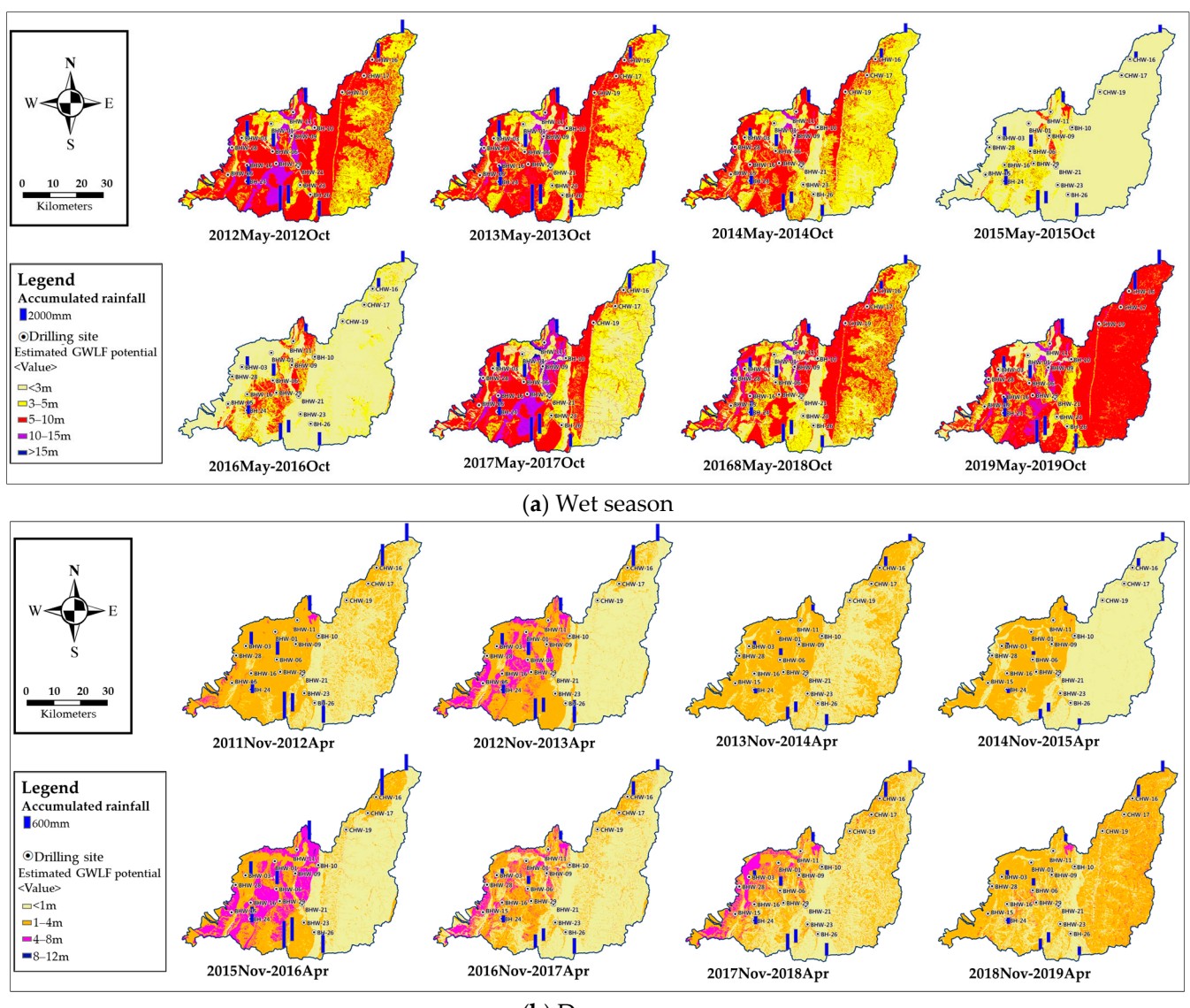

**Figure 16.** Spatial distribution of accumulated rainfall and estimated GWLF potential during 2011–2019 in the wet/dry season.

In addition, the time required for the groundwater level to return to the pre-rainfall level varies for each monitoring well. The influencing factors control the response to groundwater level changes. Taking BHW-09 as an example (Figure 17), the groundwater level variation is close to rainfall changes. However, the groundwater level is obviously delayed by the actual changes. The delayed situation may be due to the process of rainwater infiltrating into the saturated aquifer, which can be affected by the geomorphological characteristics (slope, drainage density, and land use), geological characteristics (lithology and regolith thickness), and hydrogeological characteristics (hydraulic conductivity, porosity, and depth to the water table). The fast response time of this lag situation is within a few minutes to a few hours, and the slow response may take several days [16].

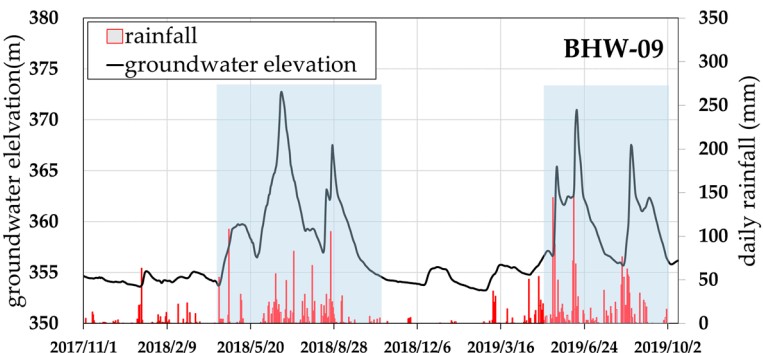

**Figure 17.** Variation in daily rainfall and groundwater elevation from November 2017 to October 2019 in borehole BHW-09.

## 5. Conclusions

Investigating the spatial and temporal dynamic behavior of the groundwater level fluctuation in mountainous areas can assist in obtaining effective strategies for developing and managing groundwater resources. This study used long-term groundwater level monitoring data in the mountainous area of Taiwan to analyze GWLF potentials during the wet and dry seasons and developed a technique for mapping the spatial distribution of GWLF potential. The main findings are summarized as follows:

1.  This study analyzed the groundwater level data from 18 monitoring stations for eight hydrological years. In the dry season, the groundwater level changed from 0.45 m to 6.35 m. In the wet season, the groundwater level changed from 1.30 m to 18.67 m. In addition, the spatial variations in the groundwater levels may not be consistent from year to year. This exception is presumed to be related to the lateral recharge behavior in the mountain area. However, this point should be carefully inspected by comparing the lateral flow data.

2.  The eight proposed environmental influencing factors, including slope, drainage density, land use, lithology, hydraulic conductivity, porosity, groundwater depth, and regolith thickness, affect GWLF potential. The contribution of each factor to GWLF potential is adjusted according to the amount of rainfall (or groundwater recharge), which is a rather complex mechanism. Although the types of data collected so far do not fully reveal this mechanism, the main controlling factors affecting groundwater level fluctuations in wet and dry seasons can be roughly obtained. In the wet season, the factors that significantly influence GWLF include K, GW, LT, D, and RT. In the dry season, the factors that significantly influence GWLF include LT, S, and P. Thus, the dominant controlling factors are not precisely the same between the wet and dry seasons.

3.  To verify the accuracy of the estimated GWLF results, the simulated results were compared with the observed GWLF from 18 groundwater monitoring wells with eight years of data. Overall, the verification results from 144 measurements demonstrate that the developed model can be expected to predict the GWLF for different seasons with a "good" level of accuracy, based on Donigian's [32] guidelines of error for calibration/verification tolerances to watershed model users. Therefore, the developed model can predict the spatial GWLF distribution based on the groundwater level data from a few wells. However, it should be noted that the proposed model needs to collect more data and other types of data (e.g., lateral discharge measurements) to improve its prediction ability.

4.  By comparing the spatial distribution of rainfall with the GWLF data, the groundwater level changes with the seasonal rainfall for most of the wells, which can lead to the inference that the groundwater circulation is pronounced. However, the groundwater level will soon return to its normal groundwater level while rainfall stops. This implies that the aquifer cannot store groundwater easily in the mountainous area of Taiwan. In addition, the time required for the groundwater level to return to the

5.      pre-rainfall level may take a few minutes to several days. The difference may be due to the rainwater infiltrating into the saturated aquifer, which can be affected by the geomorphological, geological, and hydrogeological characteristics.

5.      The core of the proposed method is to obtain the weighting coefficients of the influencing factors affecting the GWLF potential by using the in situ hydrogeological test data and groundwater level data. Compared to the previous methods of using only static hydrogeological data and constant weights, the newly developed method further strengthens the input of the dynamic behavior of groundwater into the estimation of groundwater level fluctuation potential. It shows that the estimation results will have the characteristics of dynamic changes with different rainfall conditions.

Finally, the methods of estimating the regolith thickness, water level depth, and gridded data generation should be considered under the constraints of limited field survey data. These produced data may be associated with uncertainty in predicting the groundwater level fluctuation potential. More attention must be paid to collecting more field data required for constructing reasonable and reliable prediction models.

**Author Contributions:** N.-C.C. developed the conceptualization, processed the GIS data, and wrote part of the manuscript; H.-Y.W. worked on the influencing factor preparation and processed the GIS data; C.-C.K. revised the manuscript; F.-M.L. analyzed the groundwater level data and curated the data; S.-M.H. was involved in preparing the original draft, writing, reviewing, and editing. Y.-T.L. and C.-C.H. were involved in preparing the raw data. All authors have read and agreed to the published version of the manuscript.

**Funding:** This research was supported by the Central Geological Survey (CGS), Ministry of Economic Affairs (MOEA), Taiwan (109-5226904000-01-01-01), and SINOTECH Engineering Consultants, Inc., Taiwan (RG11301).

**Institutional Review Board Statement:** Not applicable.

**Informed Consent Statement:** Not applicable.

**Data Availability Statement:** Not applicable.

**Acknowledgments:** The authors are grateful to the CGS, SINOTECH, and National Cheng Kung University for their cooperation with the projects of groundwater resource investigation in mountainous regions.

**Conflicts of Interest:** The authors declare no conflict of interest.

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
