# Peer review of "Investigation and Estimation of Groundwater Level Fluctuation Potential: A Case Study in the Pei-Kang River Basin and Chou-Shui River Basin of the Taiwan Mountainous Region"

_applsci, doi:10.3390/app12147060_

Round 1

Reviewer 1 Report

General comment:

The paper entitled “Investigation and Estimation of Groundwater Level Fluctuation Potential: A Case Study in Pei-Kang River Basin and Chou Shui River Basin of Taiwan Mountainous Region” by Chen et al., highlights factor controlling groundwater resource protentional in mountainous region of Taiwan. The ideas are efficiently presented in the discussion and conclusion section. One major comment is I have is that, it will be great if the authors releases the groundwater level data as supplementary to support the reproducibility of the work. One of my concern is the method of estimating the regolith thickness, water level depth and the gridded data generation and associated uncertainty. I recommend the author to highlight those points.

Specific comments on the paper is presented below.

Line 20: remove the full stope just before have.

Line 10: I think it is important to present as few and important abbreviation as possible, for reader it will be better to present the exact word than the abbreviation. (e.g. GWLF)

Line 40: “Much of the groundwater water, either recent anthropocene groundwater (ref.1) or older fossil groundwater (ref.2), stored under fractured rocks and sediments are vulnerable to water quality deterioration and groundwater depletion at geographically varying locations.” Please use this recent large-scale groundwater papers at geographically varying regions across the USA, it might strengthen your research question.

ref. 1 Jurgens, B. C., Faulkner, K., McMahon, P. B., Hunt, A. G., Casile, G., Young, M. B., & Belitz, K. (2022). Over a third of groundwater in USA public-supply aquifers is Anthropocene-age and susceptible to surface contamination. Communications Earth & Environment3(1), 1-9. [https://www.nature.com/articles/s43247-022-00473-y#Abs1]

ref. 2 GebreEgziabher, M., Jasechko, S., & Perrone, D. (2022). Widespread and increased drilling of wells into fossil aquifers in the USA. Nature communications13(1), 1-12. [https://www.nature.com/articles/s41467-022-29678-7]  

Line 111: Figure-1, excellent map representation and handy to identify different aquifer Hydrostratigraphic units and I suggest to make consistent font for the map layer and your text. (e.g. the UTM font size). (For all Figure with spatial map, please cross-check the font size of the main text, coordinate and legend or any text in the figure to match with your paper font type, if not size, at least.

Line 204: Section “Depth to the water table”, I like the map you made for this section (Fig. 9). To have better understanding to your reader and to present the quality of your data, I suggest to add the location of the bore holes, like a point borehole location, that has been used to produce the water level map.

Line 221: Section “Regolith thickness”, do you have any verification to support the empirical regolith thickness estimation. If no, please add this as limitation of the work. Consider this as a factor in your discussion and conclusion.

Line 248: Why is your paper repeating the figures again in Figure 11?

Line 251: On Section 3.3. “Determination of the weightings of individual influencing factors from groundwater level fluctuation data” on your assumption, how do you find the spatially gridded data relationship with each controlling factor, for example are you considering linear relationship or complex as the geology and groundwater dynamics are really complex down in the aquifer.

Author Response

Response to Reviewer 1 Comments

The authors appreciate the positive comments from the reviewer. We have reviewed the constructive comments very carefully and tried our best to revise the manuscript accordingly.

Therefore, we have prepared the detailed point-by-point response to the reviewers' comments in the fallowing text. We also provide a revised manuscript with all changes clearly highlighted using the "Track Changes" function in Microsoft Word. We hope that the corrections will meet with approval.

Point 1: The paper entitled “Investigation and Estimation of Groundwater Level Fluctuation Potential: A Case Study in Pei-Kang River Basin and Chou Shui River Basin of Taiwan Mountainous Region” by Chen et al., highlights factor controlling groundwater resource protentional in mountainous region of Taiwan. The ideas are efficiently presented in the discussion and conclusion section. One major comment is I have is that, it will be great if the authors releases the groundwater level data as supplementary to support the reproducibility of the work. One of my concern is the method of estimating the regolith thickness, water level depth and the gridded data generation and associated uncertainty. I recommend the author to highlight those points.

Response 1: The ownership of the groundwater level data belongs to the Central Geological Survey of Taiwan, and the authors have no right to release the data. In addition, the concerns from the reviewer have been added in the section of “Conclusion”.

Point 2: Line 20: remove the full stope just before have.

Response 2: The typo has been revised.

Point 3: Line 10: I think it is important to present as few and important abbreviation as possible, for reader it will be better to present the exact word than the abbreviation. (e.g. GWLF)

Response 3: Line 10 is regarding “Correspondence: [email protected] ; Tel.: +886-2-2462-2192 #6171”. I guess the reviewer may mention about many abbreviations (GWLF) showing in “Abstract”. If so, the use for the abbreviation of GWLF is due to the rule of “A single paragraph of about 200 words maximum” in Abstract.

Point 4: Line 40: “Much of the groundwater water, either recent anthropocene groundwater (ref.1) or older fossil groundwater (ref.2), stored under fractured rocks and sediments are vulnerable to water quality deterioration and groundwater depletion at geographically varying locations.” Please use this recent large-scale groundwater papers at geographically varying regions across the USA, it might strengthen your research question.

ref. 1 Jurgens, B. C., Faulkner, K., McMahon, P. B., Hunt, A. G., Casile, G., Young, M. B., & Belitz, K. (2022). Over a third of groundwater in USA public-supply aquifers is Anthropocene-age and susceptible to surface contamination. Communications Earth & Environment, 3(1), 1-9. [https://www.nature.com/articles/s43247-022-00473-y#Abs1]

ref. 2 GebreEgziabher, M., Jasechko, S., & Perrone, D. (2022). Widespread and increased drilling of wells into fossil aquifers in the USA. Nature communications, 13(1), 1-12. [https://www.nature.com/articles/s41467-022-29678-7] 

Response 4: Thanks for the suggestion by the reviewer. The authors have added two references and the issue in the text.

Point 5: Line 111: Figure-1, excellent map representation and handy to identify different aquifer Hydrostratigraphic units and I suggest to make consistent font for the map layer and your text. (e.g. the UTM font size). (For all Figure with spatial map, please cross-check the font size of the main text, coordinate and legend or any text in the figure to match with your paper font type, if not size, at least.

Response 5: The authors have made consistent font for the map layer and the text. The authors have modified all graphs.

Point 6: Line 204: Section “Depth to the water table”, I like the map you made for this section (Fig. 9). To have better understanding to your reader and to present the quality of your data, I suggest to add the location of the bore holes, like a point borehole location, that has been used to produce the water level map.

Response 6: The authors have added the borehole locations in Figure 9.

Point 7: Line 221: Section “Regolith thickness”, do you have any verification to support the empirical regolith thickness estimation. If no, please add this as limitation of the work. Consider this as a factor in your discussion and conclusion.

Response 7: The accuracy of predicted regolith thickness relies on the coefficient of determination of 0.75 for the regression model. As the suggestion by the reviewer, the limitation of this work has been added in the section of “Conclusion”.

Point 8: Line 248: Why is your paper repeating the figures again in Figure 11?

Response 8: The sub-figures in Figure 11 are different from previous figures (Figure 3 to Figure 10).  Figure 11 depicts the spatial distribution of the “feature score”, which is assigned by Table 1, for each influencing factor. However, previous figures just illustrate the spatial distribution of “raw data” for each controlling factor.

Point 9: Line 251: On Section 3.3. “Determination of the weightings of individual influencing factors from groundwater level fluctuation data” on your assumption, how do you find the spatially gridded data relationship with each controlling factor, for example are you considering linear relationship or complex as the geology and groundwater dynamics are really complex down in the aquifer.

Response 9: In this study, the selected influencing factors can be attributed as geomorphologic, geological, and hydrogeological factors (as shown in Table 1).

  1. Geomorphologic factors: The spatial distributions of the slope and drainage density factors were calculated by using 20 m × 20 m DTM (Digital Terran Model). The spatial distribution of the land use factor was generated by the use of the existing land use map produced by the survey and assessment of groundwater resources in the middle mountainous region of Taiwan [8].
  2. Geological factors: The spatial distribution of the lithology factor was produced based on the official geological map issued by the Taiwan Central Geological Survey [8]. The spatial distribution of the regolith thickness factor was constructed from 18 borehole logs survey data by performing regression analysis. The regression equation (Equation (3)), which is based on two factors of slope (S) and groundwater elevation (GE), was used for the spatial distribution of regolith thickness.
  3. Hydrogeological factors: Hydraulic conductivity, porosity, and the depth to the water table are all pointwise data obtained from the borehole-related tests and groundwater level observations. The field test data of hydraulic conductivity (double packer hydraulic test) and porosity (sonic logging) integrated with the existing hydrogeological map [8] were used to determine the spatial distribution of hydraulic conductivity and porosity, respectively. The depth to water table is related to the surface elevation, so the spatial distribution of the depth to water table was determined by the well-established relationship equation (Equation 2).

Finally, the feature scores of the eight influencing factors and their weighting coefficients were linearly superimposed to estimate the potential change of groundwater level.

Reviewer 2 Report

This paper is a well-written paper on groundwater level fluctuations. Before publishing this paper, some minor corrections are required as follows. 

Line 155~157 : These sentences are similar to the sentences in line 160~163. Remove one of them.

Line 266 and 297 : “, as shown in Figure 11”. Isn’t is Figure 12 instead of Figure 11?

Author Response

Response to Reviewer 2 Comments

The authors appreciate the positive comments from the reviewer. We have reviewed the constructive comments very carefully and tried our best to revise the manuscript accordingly.

Therefore, we have prepared the detailed point-by-point response to the reviewers' comments in the fallowing text. We also provide a revised manuscript with all changes clearly highlighted using the "Track Changes" function in Microsoft Word. We hope that the corrections will meet with approval.

Point 1: Line 155~157 : These sentences are similar to the sentences in line 160~163. Remove one of them. 

Response 1: The sentence in line 160-163 has been removed.

Point 2: Line 266 and 297 : “, as shown in Figure 11”. Isn’t is Figure 12 instead of Figure 11?

Response 2: This typo has been corrected.

Round 2

Reviewer 1 Report

Thank you for taking your time to respond my comments. I have one minor comment 

Line 42, Please modify "stored under fractured rocks" with "stored under porous and fractured rocks". 

Author Response

Comment 01: Line 42, Please modify "stored under fractured rocks" with "stored under porous and fractured rocks".

Response: The authors have modified the sentence followed by reviewer's suggestion.